# TerraGen: A Unified Multi-Task Layout Generation Framework for Remote Sensing Data Augmentation

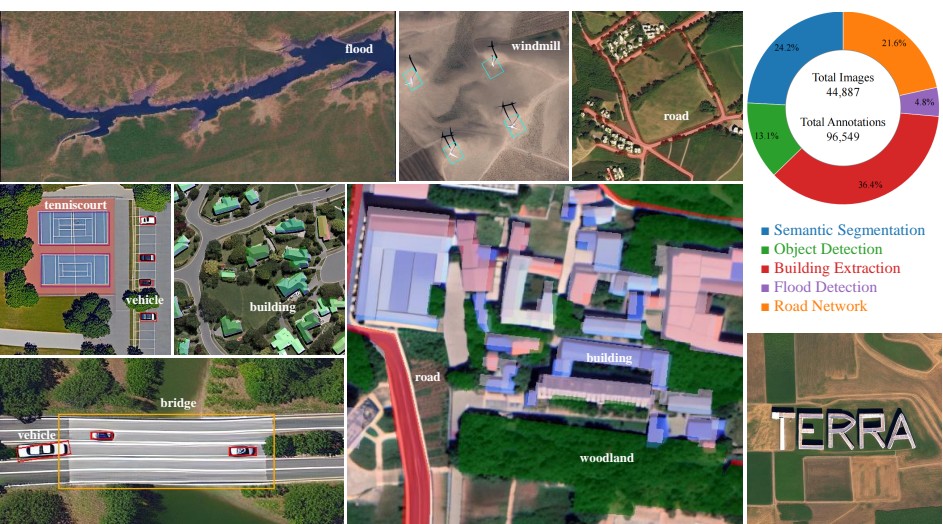

Figure 1: Generation results of the TerraGen model and composition of the TerraGen dataset. The main visuals display high-fidelity remote sensing images synthesized by our TerraGen model from given layout conditions. The pie chart in the top-right corner illustrates the five-task distribution of our purpose-built TerraGen dataset.

## Abstract

Remote sensing vision tasks require extensive labeled data across multiple, interconnected domains. However, current generative data augmentation frameworks are task-isolated, i.e., each vision task requires training an independent generative model, and ignores the modeling of geographical information and spatial constraints. To address these issues, we propose **TerraGen**, a unified layout-to-image generation framework that enables flexible, spatially controllable synthesis of remote sensing imagery for various high-level vision tasks, e.g., detection, segmentation, and extraction. Specifically, TerraGen introduces a geographic-spatial layout encoder that unifies bounding box and segmentation mask inputs, combined with a multi-scale injection scheme and mask-weighted loss to explicitly encode spatial constraints, from global structures to fine details. Also, we construct the first large-scale multi-task remote sensing layout generation dataset containing 45k images and establish a standardized evaluation protocol for this task. Experimental results show that our TerraGen can achieve the best generation image quality across diverse tasks. Additionally, TerraGen can be used as a universal data-augmentation generator, enhancing downstream task performance significantly and demonstrating robust cross-task generalisation in both full-data and few-shot scenarios.

## 1 INTRODUCTION

Deep learning has revolutionized computer vision, yet its success is fundamentally predicated upon large-scale annotated datasets. This reliance poses a significant bottleneck in remote sensing, where data acquisition is costly and annotation demands specialized expertise across diverse tasks like object detection, segmentation, and change detection. Remote sensing tasks exhibit strong spatial and semantic correlations, sharing common geographic layouts. However, current research paradigms are hampered by redundant annotation and isolated task pipelines. A single scene often requires distinct labels for each task (e.g., bounding boxes and masks), while data augmentation techniques seldom exploit inter-task consistency.

The advent of controllable generative models, such as GLIGEN (Li et al., 2023) and InstanceDiffusion (Wang et al., 2024), has established powerful paradigms for image synthesis conditioned on text and layout. While effective for natural images, their application in remote sensing remains confined to single-task or single-condition generation, thus lacking flexible multimodal control and cross-task adaptability.

More critically, remote sensing generative models (Zhang et al., 2024b; Zang et al., 2025; Toker et al., 2024; Tang et al., 2025) lack standardized evaluation benchmarks. Existing approaches are typically designed for specific tasks without unified multi-task evaluation frameworks, hindering both technical progress and practical adoption. This limitation is compounded by the unique geographic constraints in remote sensing imagery—such as road network connectivity, building arrangement patterns, and spatial relationships between land cover types—that are poorly captured by general-purpose generation methods.

The central premise of this paper is that the challenge of task isolation can be overcome by identifying a universal medium. We argue that spatial layout information serves as this universal representation, bridging different remote sensing vision tasks, while multimodal conditional control enhances generation flexibility and precision. This premise is built on three key observations: First, spatial representations across tasks (bounding boxes in detection, pixel masks in segmentation, contours in instance segmentation) fundamentally describe the same geographic object distribution patterns with shared geometric and semantic foundations. Second, geographic object layouts follow specific spatial rules that can be effectively encoded through structured representations. Third, textual descriptions provide semantic details that complement layout conditions, enabling more precise and diverse generation control, as shown in Figure 1.

Based on these insights, we introduce **TerraGen** (from the Latin 'Terra' for Earth), a multi-conditional generation framework for remote sensing imagery. Our contributions are threefold:

- We construct the first large-scale multi-task remote sensing layout generation dataset and establish standardized evaluation protocols, addressing the critical lack of unified benchmarks in this field.
- We propose a unified multi-conditional layout generation framework (TerraGen) that integrates spatial layout information (bounding boxes, segmentation masks) with semantic textual information through geographic spatial-aware conditional encoding mechanisms, achieving fine-grained generation control of complex remote sensing scenes.
- Our TerraGen can serve as a universal data-augmentation engine, markedly boosting downstream-task accuracy and exhibiting strong generalisation across tasks under both full-data and few-shot scenarios.

## 2 RELATED WORK

### 2.1 TEXT-TO-IMAGE GENERATION

Text-to-image generation has evolved from GAN-based models (Reed et al., 2016) to diffusion models (Ramesh et al., 2021; 2022; Saharia et al., 2022; Nichol et al., 2021), offering improved stability and visual quality. Recent methods (Radford et al., 2021; Podell et al., 2023; Esser et al., 2024; Labs et al., 2025) adopt Multimodal Diffusion Transformers (MM-DiT), treating text as an independent modality and enhancing synthesis via multimodal attention. However, they underperform on remote sensing data due to its unique structures and patterns (Tang et al., 2024).

## 2.2 LAYOUT-TO-IMAGE GENERATION

Layout-to-image generation extends synthesis by introducing fine-grained spatial control, typically through structured inputs like bounding boxes or segmentation masks. In the natural image domain, models like GLIGEN (Li et al., 2023) and LayoutDiffusion (Zheng et al., 2023) achieve impressive results by injecting spatial guidance into the cross-attention layers of pre-trained diffusion models. Recent MM-DiT-based extensions (Zhang et al., 2024a; 2025b;a) further advance this by integrating layout as another input modality, demonstrating strong layout fidelity. Nonetheless, these methods often rely on massive, web-scale datasets (e.g., COCO, LVIS) and still struggle with pixel-level precision for complex, overlapping instances.

In the remote sensing domain, layout-conditioned generation is an emerging but critical area. Early methods like CC-Diff (Zhang et al., 2024b) and AeroGen (Tang et al., 2025) have focused on bounding box-guided generation for object detection, while others like SatSynth (Toker et al., 2024) have explored mask-guided synthesis for segmentation. Despite their progress, these approaches are often task-specific and tend to overlook crucial domain-specific constraints. For instance, they may generate objects that are spatially plausible in isolation but violate geographic rules, such as disconnected road networks, randomly oriented buildings, or illogical land-use adjacencies. Our framework moves beyond this by proposing a unified model that handles multiple layout types and is aware of these geospatial relationships.

## 2.3 GENERATIVE AUGMENTATION IN REMOTE SENSING

The high cost of expert annotation and the inherent long-tail distribution of objects in remote sensing have motivated the use of generative models for data augmentation. Unlike traditional augmentation (e.g., rotation, scaling), diffusion-based methods can create novel, diverse, and highly realistic image-label pairs. Early successes like DiffuSat (Khanna et al., 2023) and CRS-Diff (Tang et al., 2024) demonstrated the potential of conditional generation to boost the performance of downstream segmentation models. SatSynth (Toker et al., 2024) further validated this approach for both segmentation and detection tasks.

However, a significant limitation of existing work is its fragmented, task-specific nature. Current pipelines require training separate generative models for detection, segmentation, or other tasks, creating data silos and preventing knowledge transfer. This is inefficient and fails to exploit the rich, shared information across different annotation types. For example, bounding box layouts contain valuable semantic and location information that could regularize and improve a segmentation model. Our work addresses this gap by designing an end-to-end pipeline that not only performs single-task augmentation but also facilitates cross-task knowledge transfer, realizing an "annotate once, benefit multiple tasks" paradigm that is essential for scalable, real-world remote-sensing systems.

# 3 TERRAGEN

## 3.1 PROBLEM FORMULATION

Remote sensing image generation requires addressing multiple heterogeneous conditions simultaneously, where each generated image must satisfy spatial layout constraints, semantic category information, and geographic contextual descriptions. The fundamental challenge lies in accurately integrating these diverse user-specified conditions—each representing distinct aspects of remote sensing analysis requirements—into geographically consistent and visually realistic images. Current diffusion models in remote sensing primarily focus on single-task scenarios, lacking unified frameworks that can capture geographic relationships across multiple vision tasks. We formally define the multi-task layout-conditioned remote sensing image generation as:

$$I_g = f(\mathcal{T}, \mathcal{L}, \mathcal{D}), \tag{1}$$

where $I_g$ represents the generated remote sensing image, $\mathcal{T}$ specifies the target task type from our supported task set $\{\mathcal{T}_0, \mathcal{T}_1, \mathcal{T}_2, \mathcal{T}_3, \mathcal{T}_4\}$ corresponding to object detection, semantic segmentation, building extraction, Road Extraction mapping, and flood detection respectively. The layout condition $\mathcal{L}$ provides spatial constraints, while the textual description $\mathcal{D}$ offers semantic context. The

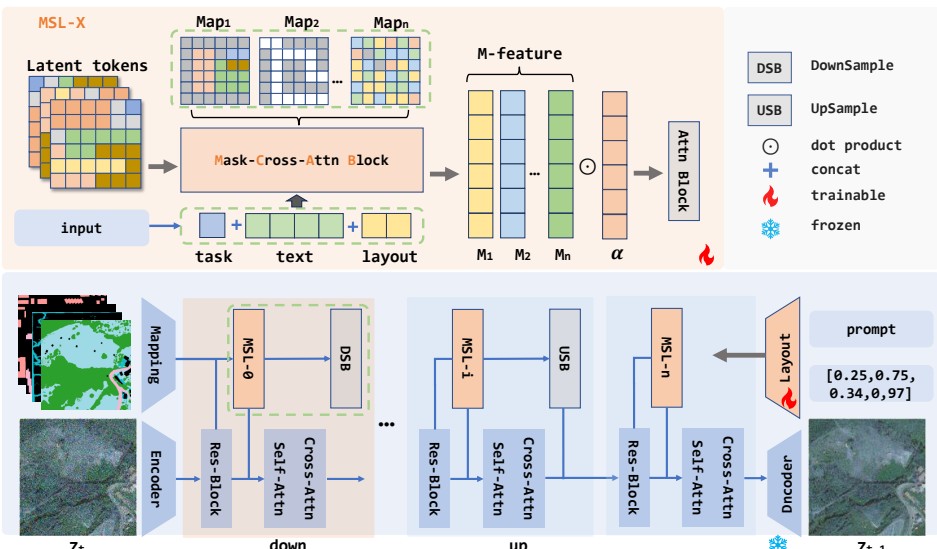

Figure 2: An overview of the TerraGen architecture. TerraGen utilizes a multi-conditional generation framework that integrates geographic-spatial layout encoding with multi-scale injection mechanisms. The system processes task specifications, textual descriptions, and spatial layouts through mask-cross-attention blocks and injects layout guidance at multiple U-Net resolution levels to ensure both global layout consistency and fine-grained spatial accuracy in RS image generation.

layout condition $\mathcal{L}$ comprises $n$ geographic entities, where each entity is characterized by a comprehensive tuple containing semantic, spatial, and attribute information:

$$\mathcal{L} = \{l_i = (c_i, s_i)\}_{i=1}^n, \tag{2}$$

Here, $c_i$ denotes the semantic category (e.g., building, road, water body), and $s_i$ represents the spatial representation, which can be either bounding boxes $\mathbf{b}_i$ for detection tasks or binary masks $\mathbf{m}_i$ for segmentation tasks. This design enables our framework to capture the complex spatial relationships inherent in remote sensing imagery.

To enable effective cross-task knowledge transfer, we establish a unified layout representation that bridges different task types through a task-adaptive transformation function:

$$\mathcal{L}_{unified} = \mathcal{J}(\mathcal{T}\mathcal{L}_{task}). \tag{3}$$

The transformation function $\mathcal{J}(\cdot)$ converts task-specific annotations—bounding boxes $\mathbf{B} = \{\mathbf{b}_i\}_{i=1}^n$ for object detection $\mathcal{T}_0$ and pixel masks $\mathbf{M} = \{\mathbf{m}_i\}_{i=1}^n$ for segmentation tasks $\mathcal{T}_{1-4}$—into a standardized layout representation while preserving essential spatial and semantic information.

## 3.2 MULTI-TASK UNIFIED ARCHITECTURE

Building upon large-scale diffusion models, TerraGen introduces a multi-conditional generation framework specifically designed for remote sensing applications. Our unified architecture effectively handles heterogeneous layout conditions while maintaining robust cross-task compatibility across diverse generation scenarios. An overview of the TerraGen architecture is shown in Figure 2, illustrating the integration of geographic-spatial layout encoding and multi-scale injection mechanisms that enable seamless task-specific generation.

To capture the unique characteristics of remote sensing imagery and achieve precise pixel-level layout guidance, we design an Instance-Spatial Layout Encoder that jointly processes multiple layout modalities through a unified embedding space:

$$\mathbf{E}_{geo} = \psi(\phi_b(\mathbf{B}), \phi_m(\mathbf{M})), \tag{4}$$

where the fusion function $\psi(\cdot)$ integrates complementary spatial representations from the bounding box encoder $\phi_b(\mathbf{B})$ and mask encoder $\phi_m(\mathbf{M})$. Here, $\mathbf{B} = \{\mathbf{b}_i\}_{i=1}^n$ represents the collection of

bounding boxes defining coarse spatial regions, and $\mathbf{M} = \{\mathbf{m}_i\}_{i=1}^n$ represents the collection of masks providing fine-grained pixel-level constraints.

The core innovation of TerraGen lies in its multi-conditional layout control mechanism that enables seamless layout-to-image generation across various remote sensing tasks. To enable task-specific generation while maintaining our unified architecture, we introduce adaptive task conditioning modules that dynamically adjust the generation process:

$$\mathbf{c}^t = \theta(\mathcal{T}) \oplus \delta(\mathbf{h}^l, \mathcal{T}). \tag{5}$$

The task encoder $\theta(\cdot)$ processes task specifications $\mathcal{T}$ to generate task-aware embeddings, while the task adapter $\delta(\cdot)$ modulates layout features $\mathbf{h}^l = \{\mathbf{h}_i^l\}_{i=1}^n$ according to specific task requirements, ensuring optimal generation quality across different remote sensing scenarios.

### 3.3 MULTI-SCALE LAYOUT INJECTION

Inspired by adapters like IP-Adapter (Ye et al., 2023) and ControlNet (Zhang et al., 2023), we inject layout conditions to guide generation. However, conventional methods suffer from progressive detail loss during downsampling, a critical flaw for pixel-level remote sensing tasks. To address this, we propose a multi-scale injection strategy that preserves fine-grained spatial information. Our strategy extends spatial attention with a hierarchical mechanism operating across multiple resolutions

$$\text{Attn}(\mathbf{Q}, \mathbf{K}, \mathbf{V}) = \sum_{k=1}^K \alpha_k \cdot \text{Softmax}\left(\frac{\mathbf{Q}\mathbf{K}^T \odot \mathbf{M}^{(k)}}{\sqrt{d}}\right)\mathbf{V}, \tag{6}$$

where the learnable weights $\alpha_k$ dynamically balance contributions across different scales, and $\mathbf{M}^{(k)}$ represents the attention mask at scale $k$ that enforces spatial constraints at the corresponding resolution level. Building upon this hierarchical attention mechanism, we inject resolution-specific layout features into each attention block of the U-Net architecture. This multi-scale injection ensures that layout guidance is maintained at every resolution level:

$$\mathbf{f}_\ell^{out} = \mathbf{f}_\ell^{in} + \zeta(\mathbf{f}_\ell^{in}, \mathbf{h}^{(\ell)}, \mathbf{M}_\ell), \tag{7}$$

where $\ell$ denotes the scale level determining the feature resolution, the cross-attention operation $\zeta(\cdot)$ facilitates seamless feature integration between layout conditions and image features, $\mathbf{h}^{(\ell)}$ provides scale-specific layout guidance extracted from our Instance-Spatial Layout Encoder, and $\mathbf{M}_\ell$ serves as the attention mask that controls spatial focus at the corresponding scale level.

This multi-scale injection mechanism enables our framework to maintain both global layout consistency and local detail accuracy, addressing the fundamental limitation of existing methods in pixel-level remote sensing image generation.

### 3.4 TRAINING AND INFERENCE STRATEGY

To provide enhanced layout-guided information injection in the training process, we introduce an adaptive mask-weighted mechanism that dynamically adjusts loss computation:

$$\mathcal{L}_{total} = \mathbb{E}_{t,\mathbf{x}_0,\boldsymbol{\epsilon}}\left[\|\boldsymbol{\epsilon} - \boldsymbol{\epsilon}_\theta(\mathbf{x}_t, t, \mathbf{c})\|^2 \odot \mathbf{W}^{adapt}\right]. \tag{8}$$

The adaptive weight matrix $\mathbf{W}^{adapt}$ is computed based on both explicit layout constraints and learned attention distributions:

$$\mathbf{W}^{adapt} = \beta \cdot \mathbf{M}_{layout} + (1 - \beta) \cdot \text{Norm}\left(\sum_{i=1}^n \mathbf{A}_i\right), \tag{9}$$

where attention maps $\mathbf{A}_i$ for layout entity $i$ are aggregated and normalized, while parameter $\beta$ controls the balance between explicit mask constraints $\mathbf{M}_{layout}$ and learned attention patterns. During training, we empirically set $\beta = 0.5$ to achieve optimal balance.

During the inference phase, we apply a unified Classifier-Free Guidance (CFG) to all input conditions, treating non-target tasks as negative samples to achieve precise and effective generation:

$$\boldsymbol{\epsilon} = \boldsymbol{\epsilon}_\theta(\mathbf{x}_t, t, \emptyset) + s \cdot (\boldsymbol{\epsilon}_\theta(\mathbf{x}_t, t, \mathbf{c}_t) - \boldsymbol{\epsilon}_\theta(\mathbf{x}_t, t, \mathbf{c}_{non})), \tag{10}$$

where $s$ is the guidance scale, $\mathbf{c}_t$ represents target task conditions, and $\mathbf{c}_{non}$ denotes non-target task conditions used as negative guidance.

## 4 DATASET AND BENCHMARK

To address the lack of unified multi-task datasets for remote sensing image generation, we construct the first dataset that supports layout-conditioned generation across five representative vision tasks: object detection, semantic segmentation, building extraction, road extraction, and flood mapping. It provides a foundation for multi-task learning and cross-task generalization.

### 4.1 DATASET CONSTRUCTION

We follow a multi-stage pipeline to ensure annotation quality and task consistency. High-resolution remote sensing images are collected from public sources (Shirshmall, 2023; Wang et al., 2021; Xia et al., 2023; Li et al., 2025; 2020; Maggiori et al., 2017; Ji et al., 2018; Gupta et al., 2019; Zhu et al., 2021; Demir et al., 2018; Mnih, 2013b). Annotations for five tasks are integrated from 12 datasets, and layout-controllable text prompts are automatically generated using the multi-modal model Qwen-VL (Bai et al., 2025).

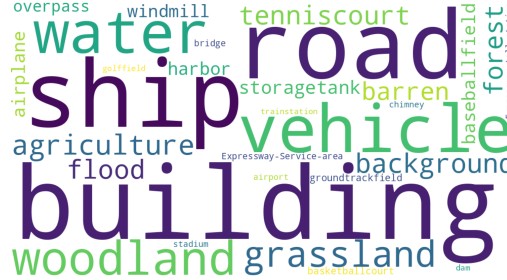

Figure 3: Word cloud of semantic categories covered in our dataset.

To enhance reliability, we adopt automatic consistency checks for detecting annotation issues such as overlapping boxes, broken road segments, and semantic conflicts. The final dataset contains 45k high-quality samples with either single-task or multi-task annotations. A word cloud in Figure 3 illustrates the rich and diverse semantic categories. Further details on our dataset construction are provided in Appendix D.1.

### 4.2 EVALUATION BENCHMARK

To establish standardized evaluation protocols for remote sensing layout generation, we constructed a comprehensive benchmark dataset specifically designed for assessing generation quality and downstream task effectiveness. Our benchmark consists of 1k carefully selected high-quality images that represent the full spectrum of challenges in remote sensing layout generation, ensuring visual complexity and geographic diversity while avoiding dense, small-scale, and low-quality samples.

Each benchmark image is associated with multiple evaluation scenarios, as a single image may correspond to generation quality assessment across different downstream tasks. To address the unique challenges of evaluating remote sensing layout generation, we introduce specialized metrics tailored for spatial accuracy and semantic consistency. These metrics encompass both pixel-level fidelity and structural coherence assessments. Detailed metric calculations are provided in the Appendix D.2.

*RS Image Quality (RS-IQ):* We calculate FID scores using an InceptionV3 network (Szegedy et al., 2016) fine-tuned on remote sensing datasets to better capture domain-specific characteristics.

*Content Fidelity:* We employ CLIP-T (Radford et al., 2021) to measure semantic consistency between generated images and global descriptions, and DINO-I (Zhang et al., 2022) to evaluate visual feature alignment.

*Layout Consistency:* We evaluate generated images using YOLOv8 (Jocher et al., 2023) -based models trained on remote sensing data, reporting mAP and $AP_{50}$ for object detection tasks, and Acc and mIoU for segmentation tasks.

## 5 EXPERIMENTS

### 5.1 EXPERIMENTAL SETUP

For model training, we adopt a two-stage configuration to ensure optimal performance and convergence. In the first stage, we train the UNet (Ronneberger et al., 2015) network with additional constraints on the diffusion process to improve the LDM (Rombach et al., 2022) -based genera-

tion capability, using a learning rate of 1e-4 with AdamW (Loshchilov, 2017) optimizer for 100k steps with batch size of 8. We employ cosine annealing scheduling to gradually reduce the learning rate throughout training. The second stage introduces an adaptive mask-weighted loss function to enhance layout consistency and spatial accuracy, incorporating attention mechanisms to better preserve structural details and geometric relationships in the generated outputs. Further implementation details and ablation studies are provided in the Appendix.

## 5.2 IMAGE QUALITY RESULTS

We compare TerraGen against both remote sensing-specific conditional generation methods and state-of-the-art natural image generation approaches. Our comparison includes remote sensing domain methods: CRS-Diff (Tang et al., 2024), SatSynth (Toker et al., 2024), AeroGen (Tang et al., 2025), and CC-Diff (Zhang et al., 2024b), as well as natural image generation advances: GLIGEN (Li et al., 2023), Uni-ControlNet (Zhao et al., 2023), ControlNet, and InstanceDiffusion (Wang et al., 2024). Due to architectural constraints and fine-tuning methodologies, we did not include DiT model (Peebles & Xie, 2023) comparisons in this evaluation.

Table 1 demonstrates TerraGen's superior performance across all evaluation metrics. Our method achieves state-of-the-art results in generation quality, Content Fidelity, and layout accuracy, establishing the effectiveness of our framework in generating geographically plausible and layout-consistent remote sensing imagery.

Figure 4 presents visual comparison results under various conditional settings. As a multi-modal controllable generation model, TerraGen demonstrates superior performance in both layout consistency and semantic alignment compared to baseline methods. For object detection tasks, our method enables precise generation of small objects such as vehicles while maintaining excellent background compatibility. For semantic segmentation tasks, TerraGen achieves better road generation quality and demonstrates fine-grained control over multiple conditions simultaneously.

| Method | Task | RS-IQ | Content Fidelity | | Mask | | BBox | |
|---|---|---|---|---|---|---|---|---|
| | | FID↓ | CLIP-T↑ | DINO-I↑ | mIoU↑ | Acc↑ | $AP_{50}$↑ | mAP↑ |
| Upper Bound (real img) | – | – | 30.8 | – | 68.6 | 83.5 | 59.5 | 42.9 |
| GLIGEN (Li et al., 2023) | OD | 42.5 | 24.8 | 53.4 | – | – | 48.7 | 33.1 |
| CC-Diff (Zhang et al., 2024b) | OD | 41.8 | 25.9 | 58.2 | – | – | 48.4 | 34.8 |
| AeroGen (Tang et al., 2025) | OD | 43.7 | 24.5 | 59.2 | – | – | 50.9 | 34.7 |
| SatSynth (Toker et al., 2024) | Seg | 49.6 | 21.1 | 48.9 | 38.2 | 58.1 | – | – |
| CRS-Diff (Tang et al., 2024) | Seg | 36.3 | 27.7 | 61.3 | 42.6 | 62.8 | – | – |
| Uni-ControlNet (Zhao et al., 2023) | Seg | 37.2 | 26.8 | 64.2 | 46.5 | 65.9 | – | – |
| InstanceDiffusion (Wang et al., 2024) | Uni | 40.1 | 25.5 | 62.1 | 41.8 | 61.3 | 51.2 | 35.8 |
| **TerraGen (Ours)** | Uni | **34.6** | **29.6** | **64.7** | **50.8** | **69.6** | **52.5** | **37.1** |

Table 1: Generation Quality and Consistency Comparison

## 5.3 DOWNSTREAM TASK ENHANCEMENT

To validate TerraGen's efficacy as a data enhancement tool, we designed three distinct experimental scenarios evaluating its performance from in-domain application to challenging data-scarce and generalization settings. In each scenario, we started with a baseline model trained on an initial dataset and then measured performance gains after enhancing the training data with synthetic images generated by TerraGen. For a detailed breakdown of the models, datasets, and training configurations used in these experiments, please see Appendix E.

**In-Domain Data Augmentation.** This experiment tested the core capability of TerraGen to improve performance when the data distribution is consistent. The baseline models were trained and tested on splits from our proposed dataset. The training set was then enhanced by adding synthetic data generated by TerraGen using the original layouts from this same dataset. As shown in Table 2, progressively adding synthetic data (from 1x to 4x the baseline quantity) yielded consistent and significant performance gains across all tasks and models. For instance, SegFormer's IoU in semantic segmentation improved by a relative 20.8%, demonstrating the direct benefit of same-distribution data enhancement.

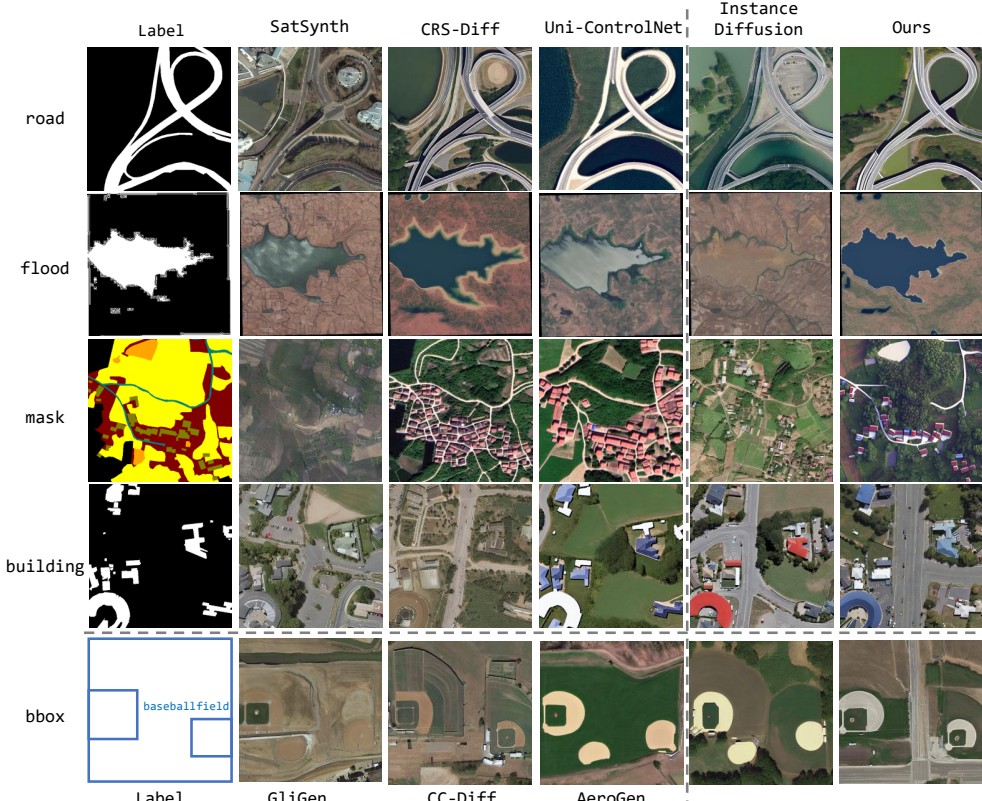

Figure 4: Qualitative comparison for mask-to-image (top four rows) and bbox-to-image (bottom row) generation by various methods.

| Task | Object Detection | | | | Semantic Segmentation | | | | Building Extraction | | | | Road Extraction | | | | Flood Detection | | | |
|---|---|---|---|---|---|---|---|---|---|---|---|---|---|---|---|---|---|---|---|---|
| Model | F-RCNN | | O-RCNN | | SegFormer | | U-Net | | SegFormer | | U-Net | | SegFormer | | U-Net | | SegFormer | | U-Net | |
| Metric | mAP↑ | AP$_{50}$↑ | mAP↑ | AP$_{50}$↑ | IoU↑ | Acc↑ | IoU↑ | Acc↑ | IoU↑ | Acc↑ | IoU↑ | Acc↑ | IoU↑ | Acc↑ | IoU↑ | Acc↑ | IoU↑ | Acc↑ | IoU↑ | Acc↑ |
| **0x** | 24.5 | 46.9 | 35.0 | 55.5 | 41.8 | 58.9 | 44.9 | 62.0 | 49.3 | 93.7 | 53.5 | 94.6 | 32.2 | 95.2 | 40.2 | 95.5 | 66.8 | 86.9 | 66.9 | 86.3 |
| **1x** | 26.2 | 49.5 | 35.1 | 57.2 | 47.3 | 64.2 | 46.1 | 63.1 | 52.7 | 94.5 | 54.9 | 95.0 | 33.8 | 95.2 | 48.1 | 96.4 | 69.1 | **87.7** | 66.3 | 86.0 |
| **2x** | 26.5 | 49.9 | 35.4 | 57.0 | 49.0 | 65.8 | 47.9 | 64.7 | 54.7 | 94.8 | 56.9 | 95.3 | 38.0 | 95.6 | 48.7 | 96.5 | 68.9 | 87.3 | 67.0 | 86.4 |
| **3x** | 27.4 | 51.1 | 35.8 | **57.6** | 50.5 | 67.1 | 47.2 | 64.1 | 55.0 | 94.9 | 56.9 | 95.4 | 38.6 | 95.7 | **49.0** | 96.5 | **69.3** | 87.5 | 67.5 | 86.5 |
| **4x** | **28.4** | **51.3** | **35.9** | 57.4 | 50.5 | 67.1 | 48.5 | 65.4 | 56.1 | 95.1 | 58.3 | 95.6 | 39.6 | 95.8 | 49.0 | 96.6 | 69.3 | 87.7 | 68.1 | 87.0 |
| vs. Baseline | +15.9% | +9.4% | +2.6% | +3.8% | +20.8% | +13.9% | +8.0% | +5.5% | +13.8% | +1.5% | +9.0% | +1.1% | +23.0% | +0.6% | +21.9% | +1.2% | +3.7% | +0.9% | +1.8% | +0.8% |

Table 2: Performance Gains from In-Domain Data Enhancement

**Few-Shot Generalization.** This experiment simulated data-scarce scenarios to evaluate TerraGen's ability to generate valuable training data when original samples are extremely limited. The baseline models were trained and tested on small, publicly available datasets (100 training samples each), completely separate from our proposed dataset. The training set was then enhanced with synthetic data generated by TerraGen. Table 3 shows that enhancing these small datasets with synthetic samples led to substantial performance improvements across all tasks, highlighting TerraGen's effectiveness in low-data regimes.

**Enhancement with Transformed Layouts.** This experiment assesses TerraGen's robustness by testing its ability to generate effective data from geometrically transformed layouts. While the baseline setup mirrored the in-domain experiment, the enhancement data was generated using layouts altered by various geometric transformations (e.g., rotation, flipping), detailed in Appendix E. The results in Table 4 confirm the efficacy of this approach, with performance gains across nearly all metrics. This is crucial, demonstrating that TerraGen robustly interprets and renders diverse spatial conditions rather than merely memorizing static layouts.

| Task | Object Detection | | | | Semantic Segmentation | | | | Building Extraction | | | | Road Extraction | | | | Flood Detection | | | |
|---|---|---|---|---|---|---|---|---|---|---|---|---|---|---|---|---|---|---|---|---|
| Dataset | NWPU VHR-10 | | | | DroneDeploy | | | | Mass. Buildings | | | | RoadNet | | | | FloodNet | | | |
| Model | F-RCNN | | O-RCNN | | SegFormer | | U-Net | | SegFormer | | U-Net | | SegFormer | | U-Net | | SegFormer | | U-Net | |
| Metric | mAP↑ | AP$_{50}$↑ | mAP↑ | AP$_{50}$↑ | IoU↑ | Acc↑ | IoU↑ | Acc↑ | IoU↑ | Acc↑ | IoU↑ | Acc↑ | IoU↑ | Acc↑ | IoU↑ | Acc↑ | IoU↑ | Acc↑ | IoU↑ | Acc↑ |
| 0x | 29.8 | 66.6 | 45.7 | 77.4 | 43.7 | 60.8 | 57.9 | 73.3 | 27.4 | 88.1 | 33.9 | 88.4 | 67.5 | 94.5 | 79.8 | 96.7 | 51.0 | 69.5 | 51.6 | **74.1** |
| 1x | 30.4 | 67.2 | 46.9 | 80.9 | 53.5 | 69.7 | 59.0 | 74.2 | 27.5 | 88.2 | 35.7 | 90.4 | 69.7 | 95.2 | 83.0 | 97.3 | 51.3 | 71.1 | 52.6 | 70.5 |
| 2x | 31.1 | 69.8 | **48.1** | 81.9 | **60.4** | **75.3** | 61.0 | 75.8 | 29.2 | 89.1 | 36.7 | 90.7 | 71.0 | 95.2 | **83.2** | **97.4** | 52.2 | 70.6 | **53.4** | 72.9 |
| 3x | **34.7** | **71.3** | 48.0 | **84.0** | 59.9 | 74.9 | **61.2** | 75.9 | **30.8** | 89.5 | 37.3 | 91.0 | 72.4 | 95.3 | 83.2 | 97.3 | 52.8 | 73.2 | 53.3 | 72.8 |
| vs. Baseline | +16.4% | +7.1% | +5.3% | +8.5% | +38.2% | +23.8% | +5.7% | +3.5% | +12.4% | +1.6% | +10.0% | +2.9% | +7.3% | +0.8% | +4.3% | +0.7% | +3.5% | +5.3% | +3.5% | -1.6% |

Table 3: Generalization Performance in Few-Shot Settings

| Task | Object Detection | | | | Semantic Segmentation | | | | Building Extraction | | | | Road Extraction | | | | Flood Detection | | | |
|---|---|---|---|---|---|---|---|---|---|---|---|---|---|---|---|---|---|---|---|---|
| Model | F-RCNN | | O-RCNN | | SegFormer | | U-Net | | SegFormer | | U-Net | | SegFormer | | U-Net | | SegFormer | | U-Net | |
| Metric | mAP↑ | AP$_{50}$↑ | mAP↑ | AP$_{50}$↑ | mIoU↑ | Acc↑ | mIoU↑ | Acc↑ | IoU↑ | Acc↑ | IoU↑ | Acc↑ | IoU↑ | Acc↑ | IoU↑ | Acc↑ | IoU↑ | Acc↑ | IoU↑ | Acc↑ |
| Baseline | 32.8 | 57.4 | 42.0 | 63.6 | 54.2 | 70.3 | 51.7 | 68.2 | 66.9 | 96.4 | 66.7 | 96.4 | 51.9 | 96.6 | 58.0 | 97.1 | 69.5 | 87.9 | 68.5 | 87.4 |
| +TerraGen | **34.0** | **57.6** | **42.6** | **64.5** | **54.3** | **70.4** | **52.5** | **68.9** | **68.2** | **96.7** | **68.0** | **96.6** | **53.4** | **96.7** | **58.2** | **97.2** | **69.7** | **88.0** | **68.6** | **87.5** |

Table 4: Enhancement with Transformed Layouts

## 5.4 ABLATION STUDIES

We conducted ablation studies to evaluate the impact of layout control, multi-scale injection, and mask-weighted loss on our model's performance. As shown in Table 5, combining both bounding boxes and masks consistently outperforms using either modality alone, improving the Fréchet Inception Distance (FID) from 38.7 to 34.6 and the mean Intersection over Union (mIoU) from 45.3 to 50.8. The half-scale multi-scale injection strategy offered a good balance between precision and computa-

| Layout Control | | MSL | MALoss | FID↓ | CLIP-T↑ | mIoU↑ |
|---|---|---|---|---|---|---|
| box | mask | | | | | |
| ✓ | ✗ | half | ✗ | 38.7 | 28.2 | 45.3 |
| ✗ | ✓ | half | ✗ | 37.1 | 28.1 | 47.8 |
| ✓ | ✓ | half | ✗ | 35.2 | 28.4 | 49.2 |
| ✓ | ✓ | full | ✗ | 35.4 | 28.3 | 49.1 |
| ✓ | ✓ | full | ✓ | 34.9 | 29.4 | 50.5 |
| ✓ | ✓ | half | ✓ | **34.6** | **29.6** | **50.8** |

Table 5: Ablation Study: Effect of Layout Control, Multi-Scale Layout (MSL), and Mask-weighted Loss (MALoss).

tional efficiency, while the mask-weighted loss further boosted all metrics, achieving the best overall performance. These results confirm the effectiveness of each component in enhancing the quality, spatial accuracy, and consistency of layout-guided remote sensing image generation.

## 6 CONCLUSION

In this work, we introduce TerraGen, a unified multi-task layout-to-image generation framework designed to overcome the critical challenges of task isolation and the absence of geographic constraints in remote sensing. Our approach features a novel geographic-spatial layout encoder that seamlessly integrates diverse spatial conditions (bounding boxes, masks) with textual descriptions, coupled with a multi-scale injection strategy for precise, controllable synthesis of spatially coherent imagery. To validate our method, we also contribute the first large-scale, multi-task remote sensing layout dataset and establish unified evaluation benchmarks. Extensive experiments demonstrate that our proposed TerraGen can achieve state-of-the-art results across various tasks, including object detection, segmentation, and flood mapping, while proving highly effective for data augmentation in both full-data and few-shot scenarios. Most notably, our framework validates the central premise that spatial layouts can serve as a universal medium to bridge previously isolated remote sensing tasks. We empirically demonstrate that this cross-task knowledge transfer is highly effective, as layouts from detection can be directly utilized to significantly enhance the performance of segmentation models, and vice versa.

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

## A   STATEMENT ON LLM USAGE

We utilized Large Language Models (LLMs), specifically Google's Gemini Pro and OpenAI's Chat-GPT, as assistants for writing and code generation during the preparation of this manuscript. Their roles included improving grammar and clarity, formatting LaTeX tables, and structuring paragraphs based on key points provided by the authors. The core scientific contributions, including all ideas, experiments, and analyses, are exclusively the work of the human authors, who take full responsibility for the final content.

## B   LIMITATIONS

While TerraGen marks a significant step forward in unified multi-task layout generation for remote sensing, certain limitations and avenues for future work exist. For instance, in object detection data synthesis, its generation quality noticeably degrades when object density becomes excessively high, for example, exceeding 20 instances in a single image. In tasks based on segmentation masks, the internal resizing to a fixed lower dimension, such as 64x64, can reduce quality for large objects or those requiring exceptionally fine-grained details. Furthermore, despite breaking task isolation and generalizing well within trained categories, TerraGen's semantic understanding is currently relatively closed-set, limiting its capacity for open-domain generation of novel or out-of-distribution objects.

## C   TERRAGEN IMPLEMENTATION DETAILS

This section provides detailed implementation specifications for the TerraGen framework, including architectural configurations, training procedures, and inference settings.

### C.1   MODEL ARCHITECTURE

TerraGen builds upon Stable Diffusion v1.5 with a UNet backbone consisting of 4 downsampling and 4 upsampling blocks. The model operates on 512×512 resolution images in the latent space using a pre-trained VAE encoder/decoder with 8× compression ratio. The layout encoder consists of two parallel branches. The *Bounding Box Encoder* $\phi_b$ employs a 2-layer MLP that processes normalized box coordinates $(x_1, y_1, x_2, y_2)$ and category embeddings to produce 768-dimensional feature vectors. The *Mask Encoder* $\phi_m$ utilizes a lightweight CNN with 4 convolutional layers, progressing through channels $1\rightarrow32\rightarrow64\rightarrow128\rightarrow768$, followed by adaptive average pooling to generate 768-dimensional mask features. The fusion function $\psi$ employs cross-attention mechanisms with 8 heads and 96-dimensional keys/queries. The final geographic embedding $\mathbf{E}_{geo}$ has dimensionality 768 to match the UNet feature channels. Layout features are injected at three resolution levels of the UNet to capture different spatial granularities. High resolution (64×64) captures fine-grained spatial details, medium resolution (32×32) handles mid-level structures, and low resolution (16×16) ensures global layout consistency. At each level $\ell$, we use specialized cross-attention blocks with learnable scale weights $\alpha_\ell$ initialized to 0.1, 0.3, and 0.6 for high, medium, and low resolutions respectively.

### C.2   TRAINING CONFIGURATION

Our training follows a carefully designed two-stage approach. In Stage 1 (Layout-Free Pre-training), we train the base UNet for text-to-image generation without layout constraints for 50k steps using learning rate 1e-4. Stage 2 (Layout-Guided Fine-tuning) introduces layout encoders and multi-scale injection with adaptive mask-weighted loss for 100k steps using learning rate 5e-5. We employ the AdamW optimizer with $\beta_1 = 0.9$, $\beta_2 = 0.999$, and weight decay 1e-2. The learning rate follows a cosine annealing schedule with warm-up for 1000 steps. Training uses batch size 8 with gradient accumulation to achieve an effective batch size of 32.

### C.3 INFERENCE CONFIGURATION

During inference, we use the DDIM scheduler with 50 sampling steps for optimal quality-speed trade-off. The guidance scale is set to $s = 5.5$ for standard generation and $s = 3.0$ for enhanced layout control scenarios. Classifier-Free Guidance is applied to all conditions, with 10% unconditional dropout during training to enable this capability. We use fixed seeds for reproducible evaluation and random seeds for data augmentation purposes. For different task types $\mathcal{T}$, we apply adaptive conditioning strategies. Object Detection tasks emphasize bounding box constraints with $\alpha_{box} = 0.8$, while Segmentation Tasks prioritize mask consistency with $\alpha_{mask} = 0.9$. Multi-Modal scenarios employ balanced attention with $\alpha_{box} = \alpha_{mask} = 0.6$.

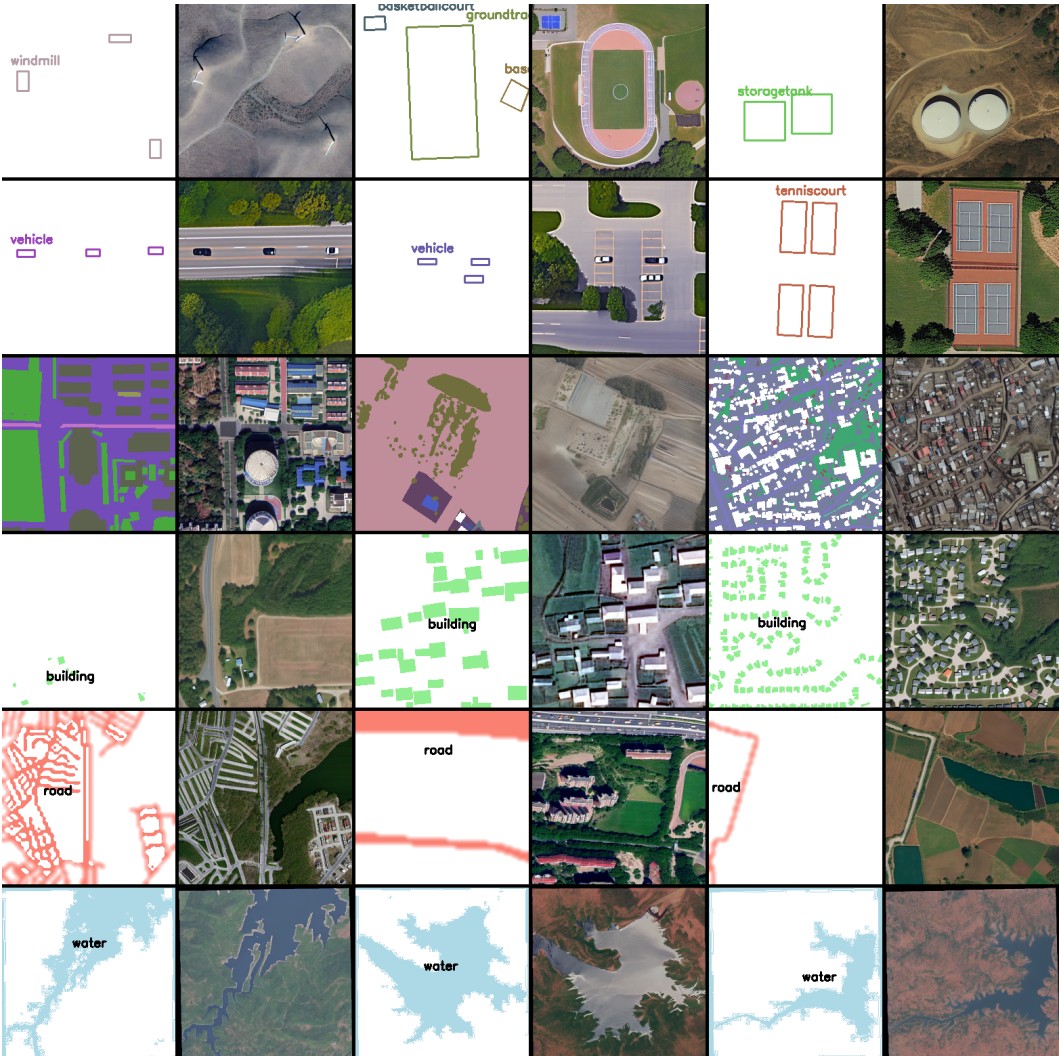

Figure 5: TerraGen Generation Showcase: Images synthesized from conditional layouts.

## D DATASET CONSTRUCTION AND EVALUATION BENCHMARK

### D.1 DATASET CONSTRUCTION

To support unified layout generation, we constructed a multi-task dataset by integrating high-resolution imagery from 12 public benchmarks (Shirshmall, 2023; Wang et al., 2021; Xia et al., 2023; Li et al., 2025; 2020; Maggiori et al., 2017; Ji et al., 2018; Gupta et al., 2019; Zhu et al., 2021; Demir et al., 2018; Mnih, 2013b).

The dataset consolidates five remote sensing tasks—Semantic Segmentation, Object Detection, Building Extraction, Flood Detection, and Road Extraction—totaling approximately 44,887 samples with 96,549 annotations across 31 categories. For controllable generation, we used Qwen-VL (Bai et al., 2025) to automatically generate text prompts for each sample. To ensure data quality, we performed automatic consistency checks to detect annotation issues like overlapping bounding boxes and broken road segments. The dataset was further refined by removing samples with overly complex annotations or poor image quality and filtering based on CLIP scores (Radford et al., 2021). This process provides a solid foundation for downstream applications. Categories for multi-class tasks are listed in Table 6, and aggregated statistics are in Table 7. In Semantic Segmentation, the unknown class is retained to improve control over unlabeled masks. The other three tasks are single-category: building, flood, and road.

| Semantic Seg. | Object Detection |
|---|---|
| agriculture | Expressway-Service-area |
| background | Expressway-toll-station |
| barren | airplane |
| building | airport |
| forest | baseballfield |
| grassland | basketballcourt |
| road | bridge |
| water | chimney |
| woodland | dam |
| unknown | golffield |
| | groundtrackfield |
| | harbor |
| | overpass |
| | ship |
| | stadium |
| | storagetank |
| | tenniscourt |
| | trainstation |
| | vehicle |
| | windmill |

Table 6: Categories for multi-class tasks in TerraGen.

| Task | Source Dataset | # Samples | # Annotations |
|---|---|---|---|
| Semantic Segmentation | LoveDA (Wang et al., 2021) (23.3%) OpenEarthMap (Xia et al., 2023) (27.7%) ReasonSeg (Li et al., 2025) (49.0%) | 10,846 | 35,797 |
| Object Detection | DIOR (Li et al., 2020) (100.0%) | 5,860 | 32,571 |
| Building Extraction | Inria (Maggiori et al., 2017) (25.6%) WHU_Aerial (Ji et al., 2018) (26.4%) WHU_SatII (Ji et al., 2018) (23.9%) xBD (Gupta et al., 2019) (24.1%) | 16,327 | 16,327 |
| Flood Detection | WBS-SI (Shirshmall, 2023) (100.0%) | 2,172 | 2,172 |
| Road Extraction | CHN6-CUG (Zhu et al., 2021) (24.3%) DeepGlobe (Demir et al., 2018) (64.3%) Massachusetts (Mnih, 2013b) (11.4%) | 9,682 | 9,682 |

Table 7: Composition of the TerraGen dataset. Percentages in parentheses indicate the proportion of samples from each source dataset.

### D.2 EVALUATION BENCHMARK

To rigorously evaluate the performance of layout generation models, we curated a benchmark test set of 1,000 samples, which is independent of the training set. These samples are sourced from the same public datasets to ensure consistent data distribution but with no overlap. The test set is composed of 400 samples for object detection, 200 for flood detection, and another 400 versatile samples designated for multi-class semantic segmentation, building extraction, and road extraction tasks. This composition guarantees comprehensive coverage of all tasks and semantic categories defined in TerraGen.

Below, we detail the calculation methods and specific experimental settings for our proposed evaluation metrics. All experiments were conducted on NVIDIA GeForce RTX 4090 GPUs.

*RS Image Quality (RS-IQ):* The Fréchet Inception Distance (FID) measures the similarity between the distribution of generated images and real images. It is calculated by fitting multivariate Gaussian distributions to the feature representations extracted by a feature network:

$$\text{FID} = ||\mu_r - \mu_g||^2 + \text{Tr}(\Sigma_r + \Sigma_g - 2(\Sigma_r \Sigma_g)^{1/2})$$

where $(\mu_r, \Sigma_r)$ and $(\mu_g, \Sigma_g)$ are the feature statistics for real and generated images. For our RS-IQ metric, we use an InceptionV3 network (Szegedy et al., 2016) whose weights were pre-trained on ImageNet and subsequently fine-tuned on a large-scale remote sensing dataset. This ensures the feature extractor is highly sensitive to domain-specific patterns. In practice, we first use our model to generate a set of images corresponding to the test prompts. Then, we use the fine-tuned InceptionV3 to extract feature vectors from both the generated images and the real ground-truth images, and finally compute the FID score.

*Content Fidelity:* This metric evaluates the alignment between the generated image and the input text prompt using two distinct scores.

- CLIP-T: This score measures semantic consistency. We utilize the official pre-trained weights of the CLIP ViT-L/14 model (Radford et al., 2021) without any fine-tuning. The computation involves a simple forward pass to extract embeddings from the generated image and the text prompt, followed by a cosine similarity calculation:

$$\text{CLIP-T} = \cos(E_I(I_{gen}), E_T(T_{prompt}))$$

- DINO-I: This score assesses high-level visual feature alignment. Similarly, we use the official pre-trained DINOv2 ViT-g/14 model (Zhang et al., 2022) without fine-tuning to extract embeddings from both the generated image and its real-world counterpart, then compute their cosine similarity:

$$\text{DINO-I} = \cos(E_D(I_{gen}), E_D(I_{real}))$$

*Layout Consistency:* This metric uses expert models trained on the TerraGen training set to evaluate layout adherence. All expert models were trained on an NVIDIA GeForce RTX 4090 GPU until convergence.

- For Object Detection, we fine-tune a YOLOv8-det model (Jocher et al., 2023) from COCO pre-trained weights. This model predicts bounding boxes on generated images, which are then evaluated using mAP and $AP_{50}$.
- For Segmentation, we train a YOLOv8-seg model (Jocher et al., 2023) on the corresponding TerraGen training splits. The model generates segmentation masks, and their accuracy is measured against the ground truth using Pixel Accuracy (Acc) and mean Intersection over Union (mIoU).

## E   DOWNSTREAM TASK ENHANCEMENTS IMPLEMENTATION DETAILS

**General Training Configuration.** All downstream models, including Faster R-CNN (Ren, 2015), Oriented R-CNN (Xie et al., 2021b), SegFormer (Xie et al., 2021a), and U-Net (Ronneberger et al., 2015), were trained on NVIDIA GeForce RTX 4090 GPUs. To ensure a fair comparison, all experiments followed a consistent protocol, which

| Object Detection | Segmentation-based Tasks |
|---|---|
| **Dataset:** NWPU VHR-10 (Cheng et al., 2014) 
 **Categories:** 
 airplane, ship, storage tank, baseball diamond, tennis court, basketball court, ground track field, harbor, bridge, vehicle | **Semantic Segmentation** 
 *Dataset:* DroneDeploy (DroneDeploy, 2019) 
 *Categories:* background, vegetation, water, building 
 **Building Extraction** 
 *Dataset:* Mass. Buildings (Mnih, 2013a) 

 **Road Extraction** 
 *Dataset:* RoadNet (Liu et al., 2018) 

 **Flood Detection** 
 *Dataset:* FloodNet (Rahnemoonfar et al., 2021) |

Table 8: Datasets and categories used in few-shot generalization experiments.

included an early stopping mechanism that halted training after 10 epochs without validation loss improvement. Each model was initialized from the same official pre-trained weights in every experimental run.

**Baseline Dataset Configurations.** We designed three distinct settings to comprehensively validate our data enhancement capabilities. The baseline sample counts for each task in each setting are detailed below:

- **In-Domain Setting:** The baseline for Object Detection utilized 2,000 samples from our proposed dataset's training split. For other tasks, the splits were smaller: 500 samples for Flood Detection, and 1,000 samples each for Semantic Segmentation, Building, and Road Extraction.
- **Few-Shot Setting:** To simulate a data-scarce environment, we used specialized public datasets with a standardized 100/50/50 train/validation/test split. The datasets and categories used are detailed in Table 8.
- **Setting for Transformed Layouts:** For the experiments involving transformed layouts, we leveraged a larger scale of our proposed dataset to establish the baselines. This involved using 5,860 samples for Object Detection, 3,500 for Semantic Segmentation, 1,000 for Flood Detection, and 8,000 each for Building and Road Extraction.

**Enhancement Data Generation.** For all scenarios, synthetic data was added in multiples (1x, 2x, etc.) of the baseline training set size.

- For **in-domain and few-shot enhancement**, synthetic images were generated using the original layouts from their respective training sets.
- For **enhancement with transformed layouts**, the layouts from our proposed dataset were first modified by a variety of geometric transformations before being used for generation. Visual examples of these transformations, which include rotation, flipping, scaling, and shearing, are shown in Figure 6.

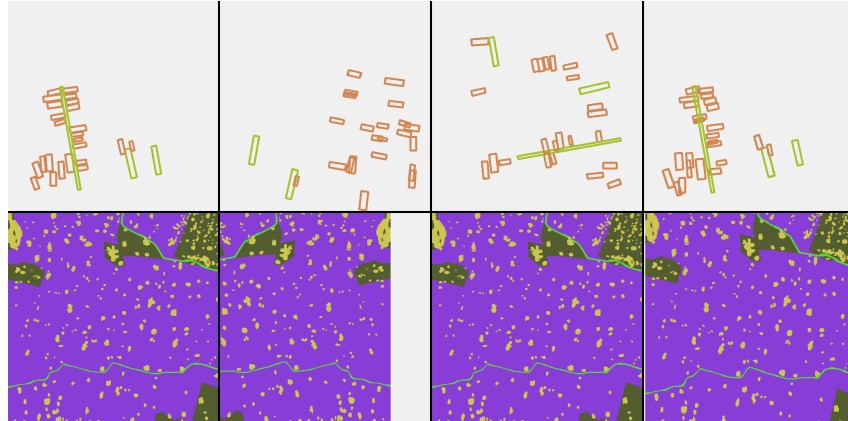

Figure 6: Visual examples of geometric transformations for data augmentation on bounding box (top row) and segmentation mask (bottom row) layouts.

