# OpenReview forum: "TerraGen: A Unified Multi-Task Layout Generation Framework for Remote Sensing Data Augmentation"
_ICLR.cc/2026/Conference — ICLR 2026 Conference Withdrawn Submission_

### Official Review · Reviewer_uSNn · 2025-10-25

**Soundness:** 2
**Presentation:** 2
**Contribution:** 2
**Rating:** 4
**Confidence:** 5

**Summary:**

This paper proposes a layout-to-image generation framework for remote sensing imagery, named TerraGen. The input could be a bounding box and a segmentation mask to generate RGB images. Additionally, this paper also constructs a multi-task remote sensing layout generation dataset.

**Strengths:**

1. A layout-to-image generation framework for remote sensing imagery is proposed,
2. A remote sensing layout generation dataset is constructed.

**Weaknesses:**

1. This paper lacks novelty. Throughout the paper, the so-called “first unified framework” essentially combines and fine-tunes existing methods (such as GLIGEN and ControlNet) for remote sensing data, lacking fundamental innovation. Most importantly, the proposed method relies solely on attention mechanisms without introducing any explicit geographical rules or knowledge, offering no fundamental improvement over existing approaches.
2. The geographic-spatial layout encoder is essentially a concatenation of the bounding box and mask encoders, with cross-attention serving as the common fusion method. The multi-scale injection mechanism represents a straightforward extension of ControlNet and IP-Adapter, lacking theoretical innovation or structural breakthroughs. Training and inference strategies (such as CFG and mask-weighted loss) are direct applications of existing techniques without adaptive modifications.
3. The baseline selection is incomplete. It does not compare with the latest DiT-based generative models (such as FLUX) nor with unified frameworks in multi-task learning.
4. The contribution of the dataset has also been overemphasized. Only 45k images should not be large-scale.

**Questions:**

Please see Weaknesses.

---

### Official Review · Reviewer_3T8D · 2025-10-31

**Soundness:** 3
**Presentation:** 2
**Contribution:** 2
**Rating:** 4
**Confidence:** 4

**Summary:**

This paper proposes TerraGen, a unified multi-task layout-to-image generation framework for remote sensing imagery. Its key contributions are: 1) a novel architecture that unifies spatial layouts with textual descriptions through geographic-aware encoding and multi-scale injection; 2) the large-scale multi-task remote sensing layout dataset with standardized evaluation benchmarks.

**Strengths:**

This paper presents a well-executed and timely study on a novel problem: unified multi-task layout generation for remote sensing data. The experimental validation is thorough and compelling, convincingly demonstrating the framework's state-of-the-art performance and its significant utility as a data augmentation engine across multiple tasks and data regimes.

**Weaknesses:**

The primary weaknesses of this paper concern the technical depth and clarity of its methodological contributions.
1.Limited Technical Innovation: While the concept of a unified multi-task framework is valuable, its core technical components, such as the layout encoder and multi-scale injection, appear to be straightforward adaptations of existing mechanisms (e.g., cross-attention, ControlNet) rather than fundamental innovations. The paper does not sufficiently justify why these specific compositions constitute a novel architectural advance beyond a direct application of established techniques to a new multi-task setting.
2.Lack of Rigor in Formulations: The mathematical presentation in Section 3 is notably lacking in rigor and clarity. Key equations are presented with undefined variables, ambiguous notations, and insufficient descriptions of their inputs and underlying mechanics. This lack of precision makes it difficult to assess, replicate, or even fully understand the proposed model, ultimately obscuring the potential technical contributions.

**Questions:**

My questions primarily seek clarification on the technical details of the model, as a response here could significantly clarify the contributions and address the noted weaknesses:
1. Regarding Equation (5): The operator ⊕ is not defined. Is this a concatenation, an addition, or another operation? The description of how this equation dynamically achieves task-specific conditioning is unclear. Could you please elaborate on the specific mechanism and the role of the task adapter δ(·)?
2. Regarding Equation (6): The inputs to the Query (Q), Key (K), and Value (V) matrices are not specified. Are they derived from the image features, the layout embeddings, or a combination? This is a critical detail for understanding the proposed hierarchical attention mechanism.
3. Regarding the scale indices: Both k in Eq. (6) and ℓ in Eq. (7) are used to denote scale levels. What is the relationship between them? Are they identical, or does k index a different set of scales within the attention block at level ℓ? This relationship should be clarified.
4. Regarding Equation (8): Several variables in the loss function are left undefined, most notably t, x_0, and ε. While these are common in diffusion models, for clarity and self-containedness, please briefly state what they represent in the context of your formulation.

---

### Official Review · Reviewer_BWtM · 2025-10-31

**Soundness:** 2
**Presentation:** 2
**Contribution:** 2
**Rating:** 6
**Confidence:** 3

**Summary:**

This paper proposes a framework for generating satellite images for data augmentation on other downstream tasks. The framework allows for the generation of the images to be conditioned on layouts, such as semantic masks or bounding boxes.

**Strengths:**

- Authors propose a multi-task unified architecture. Multi-tasks as mainly handled by converting the input conditions (bbox, segm map,...) into a common format. To differentiate between tasks, authors use a task encoder that generates task-specific embeddings
- Authors introduce a hierarchical mechanism to inject spatial information at multiple resolutions.
- Authors carry out extensive experiments, showing how TerraGen improves generation metrics compared to other models for satellite images.
- Authors demonstrate with few-shot experiments the effectiveness of the model in low-data regimes. This seems to me the main benefit of the model and encourage authors to put more emphasis on this, of its usefulness for low-data regimes.
- Ablation studies demonstrate the usefulness of the impact of the mechanisms proposed.
- Having a shared backbone between tasks improves model performance.

**Weaknesses:**

- Spelling mistakes in Figure 2: Dncoder
- Image generation is constrained to RGB images. It is worth noting that in remote sensing, satellite images have additional channel bands and wavelength frequencies. Authors should consider satellite image generation that supports the physical satellite spectrum/channel range, resulting in more physically-plausible reconstructions.

**Questions:**

- How do authors account for the different resolutions found in the pretraining dataset they use?
- How do authors ensure generated satellite image are physically plausible, and what output resolution will the generated image be? For example, authors could have used GSD Ground Sampling Distance as condition to the model
- Do authors use their model to augment other existing datasets (non few-shot generalisation setting)? I would like to know, given a model trained on a full dataset (e.g. Mass Buildings), what are the downstream improvements if we augment x2 the original Mass Buildings dataset size using TerraGen and ground truth masks? This experiment is done for in-domain setting, but it is not carried out for out-domain setting. Authors only experiment with few-shot constrained out-domain setting.

---

### Official Review · Reviewer_MEPD · 2025-11-01

**Soundness:** 2
**Presentation:** 2
**Contribution:** 2
**Rating:** 4
**Confidence:** 5

**Summary:**

This paper proposes a TerraGen, a diffusion-based model for generating remote sensing images conditioned on spatial layouts (bounding boxes, segmentation masks) and text prompts. The core of this paper is to extend the existing layout-to-image paradigm into remote sensing field.

**Strengths:**

1. TerraGen can handle multiple remote sensing tasks (object detection, segmentation, etc.) within a single model.
2. The authors constructed a dataset of 45k images with layout annotations to train and evaluate their model.
3.  Extensive experiments show that TerraGen can serve as a data augmentation engine, boosting performance on downstream tasks in both full-data and few-shot settings.

**Weaknesses:**

While the task of unified multi-task generation for remote sensing is valuable and the constructed dataset is a potential contribution, the paper has significant flaws that preclude its acceptance in its current form. My primary concerns are as follows:

1. The core technical components, i.e., a layout encoder, multi-scale feature injection, and a mask-weighted loss, are well-established adaptations of techniques from the natural image domain (e.g., GLIGEN, ControlNet, IP-Adapter). The paper presents a competent integration of these ideas but lacks a truly novel architectural or algorithmic insight. The contribution feels more like a capable engineering solution tailored for remote sensing rather than a conceptual advance in generative modeling.
2. The experimental section may not convincingly demonstrate superiority. Key baseline methods from the remote sensing domain (e.g., CRS-Diff, SatSynth) are likely designed for specific tasks and may be at a disadvantage when evaluated on a multi-task benchmark they were not designed for. A more compelling comparison would involve robustly fine-tuning these baselines on the new TerraGen dataset. The claim of outperforming natural image models is expected, given the domain shift, and does not strongly support the paper's claims.
3. The assembly of the 45k dataset from 12 different sources is a double-edged sword. The paper fails to address critical issues of domain shift, annotation inconsistency, and label noise inherent in such an amalgamation. Without detailed descriptions of the harmonization process, quality control measures, and a commitment to release the dataset, the reproducibility and generalizability of the results are severely compromised. The benchmark's validity is built upon this potentially unstable foundation.
4. Furthermore, a 45k dataset is not considered large-scale, and the author's description is exaggerated.
5. The limitations mentioned in the appendix (degradation with high object density, resizing artifacts) are significant and directly impact the practical utility of the model. However, they are only briefly acknowledged without any quantitative analysis or ablation studies to probe their causes. This gives an overly optimistic impression of the model's robustness.

**Questions:**

Please see Weaknesses

---

### Note · Authors · 2025-12-05

I have read and agree with the venue's withdrawal policy on behalf of myself and my co-authors.